# Follicle architecture and innervation of functionally distinct rat vibrissae
Ben Gerhardt [1,2,5], Tobias Rodde[1,5], Jette Alfken[3], Jakob Reichmann[3], Tim Salditt [3,6] & Michael Brecht [1,4,6] ✉

The vibrissa follicle is a complex mechanotransducer with intricate accessory structures such as vibrissa, ring sinus and ringwulst as well as rich innervation by diverse afferent types. Establishing how afferent types and accessory structures operate together to derive specific kinds of sensory information has been challenging, because we often lack precise information on afferent types, accessory structures and vibrissa function. Here we address this challenge by synchrotron X-ray imaging of vibrissa follicles of rat vibrissae with distinct function. Specifically, we characterize accessory structures and trace myelinated axons of the all-purpose-sensing C2-, an object-sensing micro-, the wind-sensing supraorbital- and the ground-sensing trident-vibrissa. We find that while vibrissa length and follicle size differ widely across these vibrissae, the ringwulst and the associated club-like afferents are of near constant diameter and height and appear to form a non-scalable sensory module. The two longer vibrissae (supraorbital and C2 vibrissa) have noticeably more club like afferents, suggesting a special role of the ringwulst in transducing presumably smaller deflection angles encountered by long sensory hairs. The trident vibrissa receives overall few afferents, which are strongly polarized to the posterior vibrissa-shaft, a putative specialization to sensing forward-egomotion. We conclude that high-resolution structural analysis allows relating follicle architecture and function.

Seminal neuroanatomical work has described the innervation and architecture of the rodent vibrissa follicle[1]. Such studies and subsequent comparative work[2] provided a rich and detailed picture of the afferent innervation. Also, early investigators characterized differences between functionally different vibrissae[3]. A problem of earlier work based on serial sections is that such work does not easily allow to trace complete axons and that it is difficult to put the intricate innervation patterns in the context of the mechano-accessory structures of the follicle.

Decisive functional information about vibrissal innervation came from afferent recording and labeling experiments. Such experiments identified club-like afferents as a set of ringwulst-associated axons that are highly sensitive and discharge at high rates[4]. By a similar methodology, Furuta et al.[5] identified Merkel-afferents as the most direction-sensitive vibrissal afferents. These identified afferent recordings complemented earlier extracellular recording work that found vibrissal afferents to discharge with great temporal precision[6,7].

Another approach to vibrissa function, i.e., the type of sensory information which individual vibrissae sample, was the study of specific, uniquely identifiable whiskers. Such an analysis was directed to the rat submandibular whisker trident, the middle one of which is the rat's only unpaired whisker[8]. The trident whisker straddles the ground during exploratory locomotion, somatosensory cortical responses carry velocity information[9], and these data suggested the trident whisker functions in egomotion-ground-sensing as a tactile speedometer.

Similarly, rat anemotaxis behavior[10], i.e., wind-sensing, has been attributed to the large supraorbital vibrissa by analog analysis of its biomechanics, somatosensory cortical responses, and behavioral blockade experiments[11].

Other studies characterized vibrissa function more broadly across rodent facial whiskers, which appear to be organized in functional groups that may derive particular features during tactile exploration[12]. In tactile object discrimination experiments by walruses[13], walruses used their smaller central whiskers rather than the larger lateral whiskers for object discrimination. Similar observations were made in rats, which also relied on the small central micro-vibrissae for fine tactile exploration[14] and tactile object recognition, whereas the larger lateral macro-vibrissae were used for

[1]Bernstein Center for Computational Neuroscience, Humboldt-Universität zu Berlin, Berlin, Germany. [2]Max Planck School of Cognition, Leipzig, Germany. [3]Institut für Röntgenphysik, Georg-August-Universität Göttingen, Göttingen, Germany. [4]Neurocure Cluster of Excellence Berlin, Berlin, Germany. [5]These authors contributed equally: Ben Gerhardt, Tobias Rodde. [6]These authors jointly supervised this work: Tim Salditt, Michael Brecht. ✉e-mail: michael.brecht@bccn-berlin.de

orienting rather than for recognition[15]. Also, other recent work[16] implicated the rat micro-vibrissae as a sensor for high-resolution tasks like texture discrimination. Similarly, seals have been shown to maximize vibrissa contacts by using the densely arranged smaller, rostral vibrissae to determine object size through tactile exploration[17]. A large body of research has investigated the broad utilities of C2 (and in general macro-vibrissae) in guiding orientation[15,18], object localization[19–21], discrimination tasks[22–24], and social interactions[25].

Most work thus far has focused on studying macro-vibrissae (i.e., C2), and the above-mentioned work on specialized vibrissae is comparably scarce. It is likely, however, that functions that have been ascribed to these vibrissae are not unique but can rather also be found in vibrissae with similar biomechanics, facial positioning, and behavioral engagement. Accordingly, the thin and dorsally positioned alpha vibrissa carries wind responses too[11] and ventrally positioned E-row whiskers also touch ground and thus also carry ground responses[9]. These effects are, albeit much stronger pronounced, in very specialized vibrissae.

Our work here builds on the aforementioned earlier research and aims at exploring the substrate for vibrissa sensing diversity (all-purpose-sensing in a C2 vibrissa, object-sensing in a micro-vibrissa, wind-sensing in the supraorbital vibrissa, and ground-sensing in the trident vibrissa). To this end, we directed the power of synchrotron X-ray phase-contrast imaging at functionally distinct rat vibrissa follicles, which provided a synoptic view of follicle afferents and accessory structures. We asked how accessory structures and the densely reconstructed afferent population compare across whiskers.

## Results

### Function and position of distinct rat vibrissae
As introduced above, our study aimed at understanding the mechanosensory underpinnings of functional vibrissa diversity. To this end, we compared four functionally distinct vibrissae. The positions of these four vibrissae are shown in a 3D rendering of a microCT scanned rat head in Fig. 1a. The vibrissae studied included: The submandibular trident (trident; red), the micro-vibrissa (micro; green), the large supraorbital vibrissa (lSO, blue), and the C2 macro-vibrissa (C2, orange). As summarized in Table 1 these vibrissae possess distinct functional characteristics. The trident-vibrissa is specialized for ground-sensing and might function as a tactile speedometer[9]. The micro-vibrissa is involved in object-sensing[14–16]. The lSO-vibrissa is decisive for wind-sensing in rats[11] and the C2-vibrissa has a broad functional range. Note that the behavioral evidence for specific vibrissae functions is rather limited, and we view all aforementioned functional assignments of the different vibrissae more as working hypotheses than ultimate functional classifications. These four vibrissa-types are positioned in distinct orientations (Fig. 1a) that support their behavioral engagement and putative function, i.e., downward in the trident (ground-sensing), frontal for the micro-vibrissa (object-sensing), upward for the lSO (wind-sensing), and lateral with wide coverage for the C2-vibrissa (all-purpose sensing). We extracted follicles of the respective vibrissae and stained them with either osmium or following electron microscopy staining protocols[26] (see methods) for X-ray phase-contrast imaging at the GINIX parallel beam setup (DESY, Hamburg). This configuration allowed the acquisition of 1.5 mm³ volumes per scan at 650 nm isotropic voxel size. Semi-transparent renderings of the obtained volume images (Fig. 1b) show differences in size and shape of the respective follicles. Follicle width and depth were less variable than height. In virtual 2D sections, X-ray dense osmium-stained myelinated axons can be recognized (Fig. 1c), along with major follicle structures.

### Variable hair length and follicle size, and near-constant-size ringwulst
Vibrissa lengths and diameter differed widely the studied vibrissae, with the micro being the shortest, followed by the medium-sized trident, long C2 and the relatively longest lSO (Fig. 2a). Three-dimensional reconstructions allowed precise visualization of afferent innervation along with accessorial

follicle architecture including ring sinus, ringwulst, hair shaft, inner conical body and outer root sheath (Fig. 2b). Reconstructed myelinated axons show a wide-ranging and highly structured innervation of the mechanosensory machinery. We normalized the positioning of these major follicle structures, which showed a rather similar relative positioning of accessory structures across the different follicles (Fig. 2c). Smaller variations could be found between the inner conical bodies of the C2, lSO and the trident and micro (Fig. 2c). Top and side view of ringwulst volume renderings shown at the same scale reveal a near-constant ringwulst size (with respect to height and diameter) across different-sized follicles (Fig. 2d, e).

In contrast, the aperture of the ringwulst shows larger differences between follicles, being widely open in trident and micro (70–80°) and nearly closed in the lSO and C2 (20–30°).

### Innervation patterns of myelinated afferents across different follicles
The reconstruction of myelinated deep vibrissal nerve axons revealed a stereotyped macroarchitecture across all four vibrissae: (i) vertical trajectories predominate, (ii) axons aggregate into axon arms[27], (iii) branching is restricted to terminal regions and (iv) axonal trajectories closely relate to accessorial structure (Fig. 3a). The axonal trees of both trident and micro appear as compressed versions of the larger and more elongated lSO and C2 innervation (Fig. 3a). In the trident and micro-vibrissa, axons are supplied as one fasciculated nerve, while the lSO and C2 follicle are supplied by two separate large nerve branches (Fig. 3b). Counts of myelinated axons revealed smaller variations for trident, micro and lSO ($n = 72 \pm 11$, $n = 87 \pm 9$, $n = 105 \pm 12$, respectively), while the numbers of C2 follicle innervation ($n = 174 \pm 2$) is fairly constant (Fig. 3c). Quantification of fiber diameter (axon plus myelin sheath) revealed significant differences between follicles, with the lSO having on average the largest fiber diameter, followed by trident and micro and smallest fiber diameter in the C2 follicle (Fig. 3d). We normalized the vertical positioning of afferents to the respective follicle heights, which showed similar patterns of distribution (Fig. 3e). However, the relative height of the follicle which receives innervation differs between follicles (Fig. 3e). Specifically, innervation of the micro-vibrissa targets relatively proximal follicle regions (50–75% follicle height), innervation of the C2 and lSO targets comparably mid follicle height (55–80% follicle height) and trident innervation relatively more distal follicle regions (60–85% follicle height). Such differences might relate to differences in biomechanics imposed by vibrissa characteristic deflection patterns of the respective follicles[28]. In all four follicles, we observed a systematic relationship between fiber diameter and terminal height (Fig. 3f). Similar to the relationship between fiber diameter and terminal height that we described for the C2-vibrissa[27], there appeared to be a proximal-to-distal axonal conduction velocity gradient along the follicle.

### Types and endings of myelinated afferents across different vibrissae
The myelinated axonal afferents of the deep vibrissal nerve divide into four main afferent types, including (1) Merkel-endings attached to the bases of the Merkel cells, (2) lanceolate-endings having a palisade-like appearance, (3) club-like-endings attaching to the upper ringwulst edge, and (4) Ruffini-endings which target the shaft at the cavernous sinus level with a trajectory towards the shaft[1,2,5]. Most Merkel- and lanceolate-endings (Fig. 4a) reach far above the ringwulst. Merkel-endings terminally penetrate the glassy membrane to attach to the bases of the Merkel cells inside the outer root sheath (Fig. S1). Lanceolate-endings lie vertically against the glassy membrane frequently with a fork-like axon ending (Fig. S1). These very fine ending morphologies could not be reconstructed in all their detail as visibility of unmyelinated axonal aspects was hampered by resolution limits. Instead, our classification[27] was based on the presence of the aforementioned morphological criteria of glassy membrane piercing in the case of Merkel-endings and terminal adjacency to the glassy membrane in case of the lanceolate-endings. Overall, distribution patterns were similar across follicles (Fig. 4a–c). Biggest differences lie in the appearance of club-like

**Fig. 1 | Overview and position of functionally distinct rat vibrissae. a** MicroCT volume rendering of an iodine-stained rat head. The functionally distinct vibrissae are highlighted in color, including the large supraorbital (lSO; blue), C2 (orange), submandibular trident (trident; red), and Microvibrissa (micro; green). **b** Semi-transparent renderings of the obtained synchrotron X-ray volumes of the respective vibrissa follicles. Follicle outlines are shown as colored shades. **c** Cross-section through the respective image volumes at the level of the nerve entrance. Boxes mark deep vibrissal nerve axons that are shown at higher magnification in the colored insets.

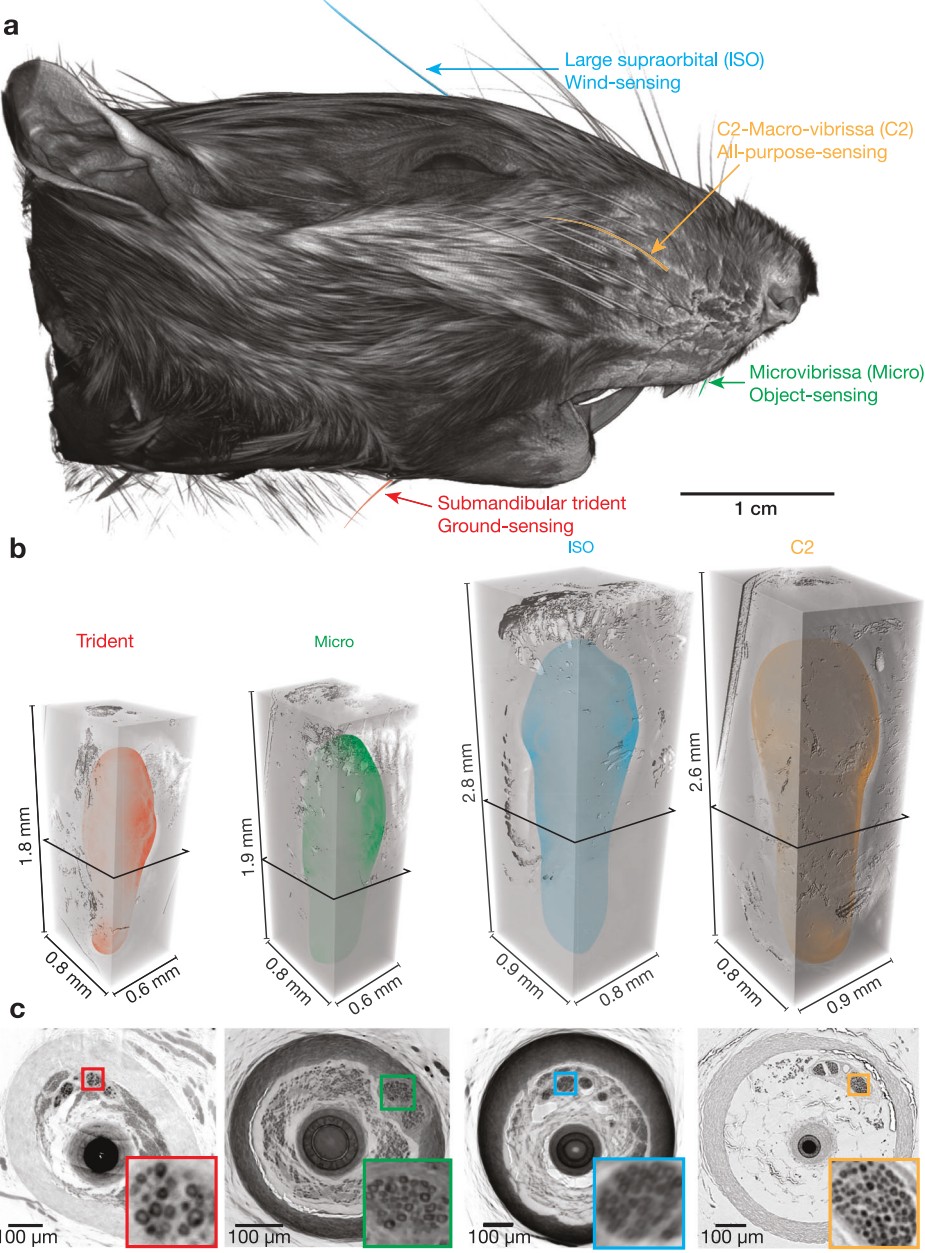

afferents, which only sparsely and heterogeneously innervate the ringwulst in the trident, micro and lSO, while the C2 ringwulst receives dense and evenly dispersed club-like innervation (Fig. 4b). Further, club-like afferents appear to innervate the ringwulst center in case of the micro, while rather innervating the ringwulst sides in the lSO and trident (Fig. 4b). Quantification of afferents types showed a near-constant amount of Ruffini-endings in all vibrissae and increasing abundance of club-like endings from trident to micro, lSO and C2 vibrissa (Fig. 4d). Merkel-endings appeared to dominate the ring sinus region in the Micro and C2 (Fig. 4d). This ratio is inverted in the trident and equal in the lSO (Fig. 4c).

**Afferent innervation relates to the ringwulst aperture**

In all four follicles, we observed asymmetric innervation patterns related to the ringwulst orientation as can be seen in top-views of axonal trees (Fig. 5a). Specifically, it appears the ringwulst aperture receives densest innervation, while the ringwulst center receives sparsest innervation. Polar histograms confirmed this impression and also showed that the magnitude of innervation polarization differs between the four follicles (Fig. 5b). Quantification of afferents present on the ringwulst aperture half of the follicle compared to

the opposite side, revealed strongest polarization in case of the trident (70%), while polarization appeared constant across micro, lSO and C2 (57–60%; Fig. 5c).

## Discussion

We observed both similarities and differences between follicles of functionally distinct rat vibrissae. As we already knew before, vibrissa follicles share the same basic accessory elements, i.e., ringwulst, inner conical body, ring sinus, and capsule. Accessory structures differed markedly in size across follicles, with the exception of the near-constant size ringwulst. Innervation differed in afferent number and spatial distribution across follicles, with a more proximal innervation observed in the micro-vibrissa.

All vibrissa follicles shared the same basic accessory structures, i.e., the basic mechanosensory layout appears to be similar across the functionally different vibrissae. Thus, all vibrissae had a capsule, a cavernous sinus, a ring sinus, and a ringwulst. The details of these mechanosensory accessories differed across vibrissae, however. The most striking differences concern vibrissa length, which correlates with follicle size. The hair length and follicle size differences we describe are in line with earlier observations[8,11,15,29]. As

**Table 1 | Location, behavioral relevance, and cortical responses of functionally distinct vibrissae**

| Vibrissa | Location | Behavior | Cortical responses | References |
|---|---|---|---|---|
| Trident | Submandibular, unpaired | Constant ground-touch; egomotion detection | Changes in locomotion speed | The et al., 2013 (Ref. 8), Chorev et al., 2016 (Ref. 9) |
| Micro | Snout proximity, dense arragement | Object detection; food-intake, foveal exploration | Surface texture differences | Brecht et al., 1994 (Ref. 15), Grant et al., 2012 (Ref. 14), Kuruppath et al., 2014 (Ref. 16) |
| ISO | Dorsally exposed, above the eyes | Wind-sensing; minimal vibrissa deflections | Wind and light stimuli | Mugnaini et al., 2023 (Ref. 11) |
| C2 | Central vibrissa pad, lateral coverage | All purpose; active touch, navigation, social touch | Object touch, whisking, navigation, discrimination | Vincent, 1912 (Ref. 18), Knutsen et al., 2006 (Ref. 21), Carvell et al., 1990 (Ref. 24) |

proposed earlier, hair biomechanics have effects on what type of sensory stimuli can be detected by a vibrissa[30], i.e., long, flexible vibrissae are uniquely suited for detecting fine movements like wind[11]. The huge follicle size differences, the near-constant size of the ringwulst of different vibrissae, are all the more surprising. This goes along with the observation of near-constant follicle diameter of vibrissae that are variable in length (verbally communicated by Hartmann and colleagues). As yet, we do not know what the meaning of this constancy is. We suggest that the ringwulst, together with the club-like afferents that innervate it and attach to it[4], forms a non-scalable sensory module.

Vibrissa follicles also shared innervation characteristics. In all follicles, the vertical innervation is distributed in similar patterns, and fiber diameter displays an increasing dependency on their vertical territory, i.e., thin afferents innervate proximal follicle regions while large afferents innervate distal follicle regions. This suggests that temporally critical transduction processes occur in distal follicle regions and that this property is shared across follicles. Further, all follicles displayed asymmetric innervation patterns, where the ringwulst aperture half of the follicle received the densest innervation, while the magnitude of this polarization varied between follicles. The numerous innervation commonalities make us wonder if these similarities reflect a basic neurogenetic *bauplan* (blueprint) of different rat vibrissa follicles.

Innervation differed strongly in the number of ascending afferents. The shorter vibrissae had fewer afferents. These trends in afferent numbers across vibrissae agree with earlier work[31]. Further, the fiber diameter of afferents differed across follicles, where Trident and micro-vibrissa show the largest fiber diameter, followed by the lSO and C2 with relatively thinnest afferents. Such differences may reflect molecular and functional sub-specializations of afferent types in different follicles with similar structures. The details of the spatial distribution of afferents also differed. The micro-vibrissa had a noticeably more proximal innervation distribution than other follicles. We expect that the micro-vibrissa will encounter relatively large deflection angles during object contacts. Along this line, we reckon the less distal innervation of the micro-vibrissa might be related to a smaller need to detect small deflections.

In all follicles, we detected the same afferent types. While there were no categorical differences in afferent types, the relative abundance of afferent types differed across follicles. The trident vibrissa had a highly polarized distribution of afferents posterior to the whisker hair. We speculate that afferent polarization might be instrumental in informing the animals about heading direction. This idea is in line with earlier biomechanical[8] and physiological work[9] that pointed to a role of the trident vibrissa in egomotion-sensing. The C2 vibrissa and lSO had relatively similar afferent compositions. Both of these vibrissae were relatively long, and we wonder if afferent similarity is related to the similarity of length.

Our work agrees in many respects with earlier work on vibrissa diversity[3]. In agreement with earlier work, we found a similarity of accessory structures, basic innervation, and afferent types. Thus, it is clear that the diverse vibrissa follicles described here compare to earlier descriptions of vibrissa follicles[1,32]. What is different in our work is the level of detail and the precise 3D registration of data provided by synchrotron imaging. While conventional microCT imaging has been instrumental in delineating macroscopic morphological features[11,33], synchrotron imaging overcomes the limitations of such laboratory-based X-ray scanning techniques posed by limited contrast and resolution. By providing micrometer resolution of mm³-sized samples, synchrotron imaging provides an equivalent to light microscopy, but extends the analysis to three dimensions. The power of this imaging also uncovered many more differences in accessory structures and afferent types between vibrissa follicles than previous work. The synoptic view of densely reconstructed follicles will therefore greatly change our thinking about vibrissa function.

## Methods
### Animal specimen
All animal specimens have their origin in animal waste from other experiments. The experiments from which we received cadavers were carried out according to German law for animal welfare and approved by the State Office for Health and Social Affairs committee (LAGeSo) in Berlin (Animal license number: G0279/18, G0095/21) and were killed according to the specific animal experimentation permits. Animals were transcardially perfused with 4 °C 0.1 M PBS, followed by 4% paraformaldehyde (PFA). Subsequently, heads were removed from the body and stored in 4% PFA at 4 °C until further use.

### Follicle extraction
The vibrissa follicles were extracted from the head of a 7-week-old male Long-Evans rat cadaver. The vibrissa follicles of the osmium and uranyl stained trident, micro, lSO, and C2 vibrissa follicle stem from one, and the only osmium-stained vibrissa follicle stem from another animal. First, the whisker pad, including the musculature, was generously removed via scalpel incisions following the border of the whisker pad. Next, the whisker pad was

**Fig. 2 | Hair length differences, different-sized follicles, and near-constant-size ringwulst of functionally distinct vibrissae. a** Silhouettes of trident, micro, lSO, and C2 vibrissa. **b** 3D segmentation of overall follicle architecture across functionally distinct vibrissae. Capsule is shown in transparent gray, ring sinus in red, inner conical body in cyan, shaft and outer root sheath in dark gray, ringwulst in yellow, and afferent innervation in multicolor (deep vibrissal innervation enters the follicle at 1/3 height and the superficial nerves enter the follicles apically). Follicles differ markedly in size. **c** Relative positioning of accessory structures along the relative follicle height. Color conventions as in (**b**). **d** Top and side views of ringwulst volume renderings shown at the same scale. Ringwulst orientation has been aligned across follicles. Note the similar size of the ringwulst across different follicles. **e** Sizes of ring sinus, ringwulst, capsule, hair, and inner conical body across follicles relative to the size of the respective structure in the trident follicle. Sizes refer to the largest diameter of the structures.

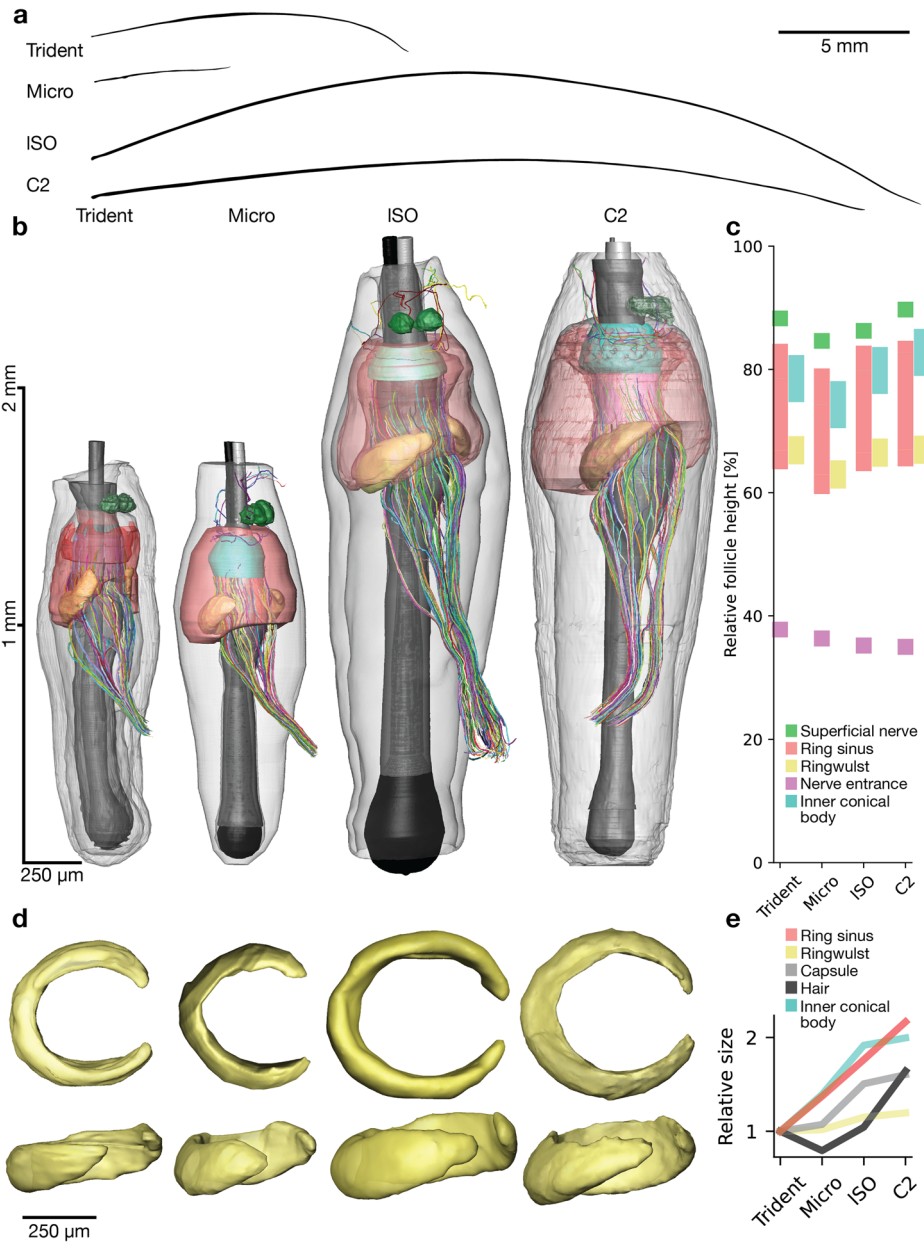

**MicroCT imaging**

The head was air-dried and glued onto the lid of a plastic container, which was mounted onto the microCT manipulator device. The MicroCT scan was acquired with a standardized YXLON FF20 CT system (YXLON International GmbH, Hamburg Germany) equipped with a Perkin Elmer Y Panel 4343 CT detector and 190 kV nano focus transmission tube. 1800 transmission images were step-wise obtained over 360° rotation, with 1 s exposure at 60 kV and 90 μA. Tomographic reconstruction was performed using the built-in YXLON reconstruction software with standardized settings.

**Sample staining**

The whole head of a 7-week-old Long-Evans rat was stained in 1% Lugol's solution (Morphisto, Cat. #10255) buffered in 0.1 M PB for 1 week at room temperature on a shaker. To remove access staining solution, the sample was washed in 0.1 M PB overnight preceding the microCT scan.

The extracted C2 and one trident vibrissa follicles was stained with 1% $OsO_4$, 0.1 M PB overnight at 4 °C. Subsequently, the sample was washed in 0.1 M PB for 24 h to remove excessive staining. $OsO_4$ is a lipophilic and X-ray dense stain that increases contrast of myelin sheaths.

The extracted trident, micro, lSO, and C2 vibrissa follicles were stained according to the Hua heavy metal staining protocol with slight modifications. Briefly: Samples were washed in 0.15 M sodium cacodylate buffer (Cac) for 1 h, followed by 2% $OsO_4$ in 0.15 M Cac, 2.5% Ferrocyanide in 0.15 M Cac for 1.5 h, $2 \times 0.5$ h wash in $H_2O$, 1% Thicarbohydrazide in $H_2O$ for 1.5 h, $2 \times 0.5$ h wash in $H_2O$, overnight incubation in 2% uranyl acetate (UA) in $H_2O$ at 4 °C, 2 h UA at 50 °C and finally another 0.5 h wash in $H_2O$. The addition of uranyl acetate in this staining protocol leads to increased contrast of intracellular components, such as proteins and nucleic acids. We provide an overview of which follicle was stained how in Supplementary Table 1.

Placed with the lateral side facing downwards, and the C2 vibrissa follicle was located. Using fine tweezers musculature and connective tissue surrounding the C2 follicle were carefully removed. The isolated follicle was then removed from the pad by small incisions in the skin.

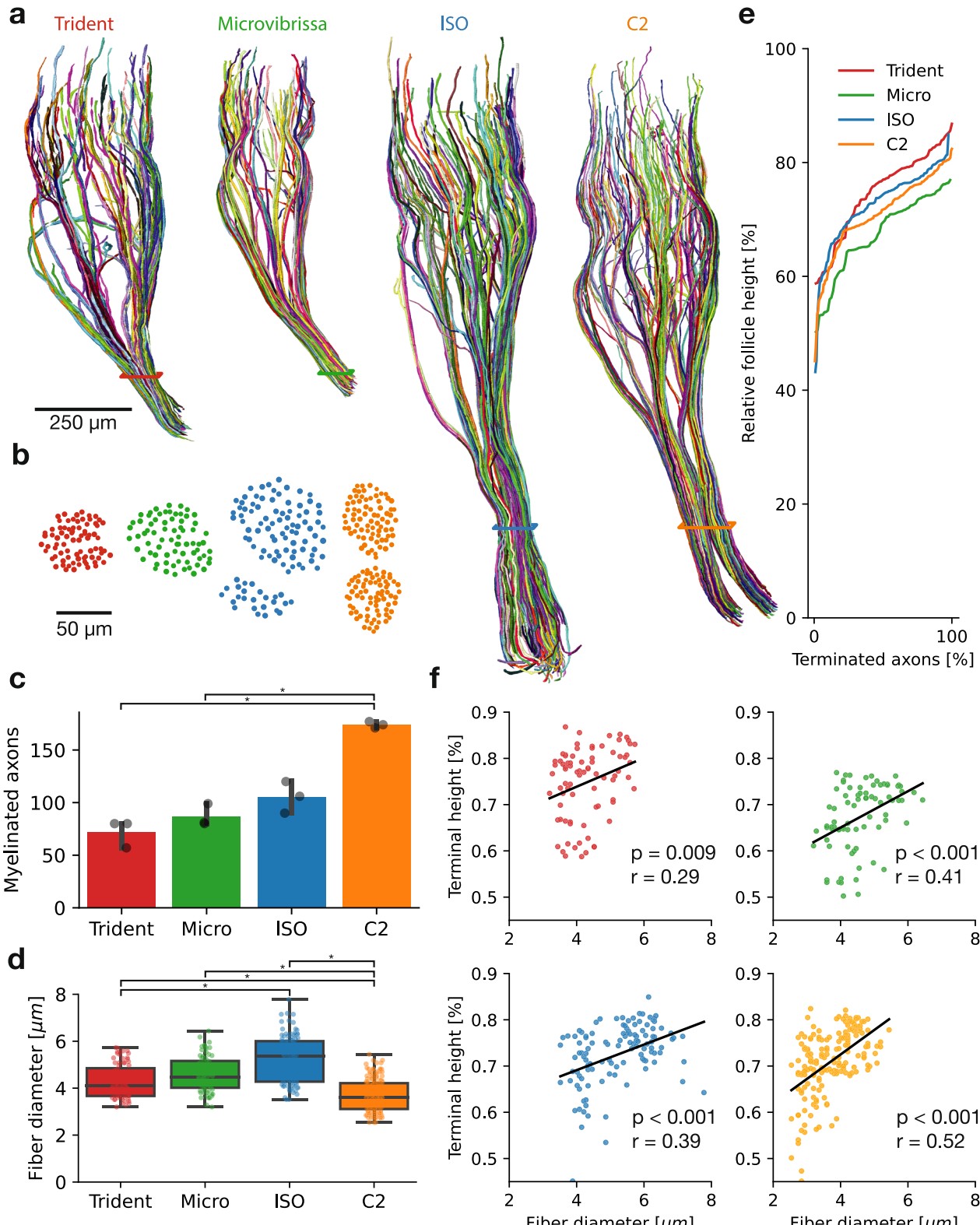

**Fig. 3 | Overview of myelinated afferent innervation patterns across different follicles. a** Volume rendering of reconstructed deep vibrissal nerve myelinated afferent axons (random multicolor assignment) per follicle. Horizontal lines around the nerve bundles indicate the plane of the nerve cross-section in (**b**). **b** Cross-section of segmented myelinated axons in the nerve entering the follicles as indicated by colored boxes in (**a**). Dot size is proportional to fiber diameter. **c** Bar plots of myelinated axon counts ($n = 3$ counts for all four follicles) (one-way ANOVA: $F = 46.33$, $p < 0.001$; significance according to Dunn's Bonferroni corrected pairwise comparison with alpha = 0.05). **d** Boxplots of fiber diameter (axon and myelin sheath) per follicle. Asterisk indicates significant differences (one-way ANOVA: $F = 88.82$, $p < 0.001$; significance according to Dunn's Bonferroni corrected pairwise comparison with alpha = 0.05). Boxplots display the 25th to 75th percentile range as the box and the median as the center line. Boxplot whiskers extend by the interquartile range. Dots show individual data points. **e** Cumulative distribution of axon endings along the relative follicle height (%). **f** Fiber diameter plotted against relative terminal height in the follicle across the four vibrissae. Color conventions as in (**a**–**e**). *P* and *r* value refer to linear regression (black line).

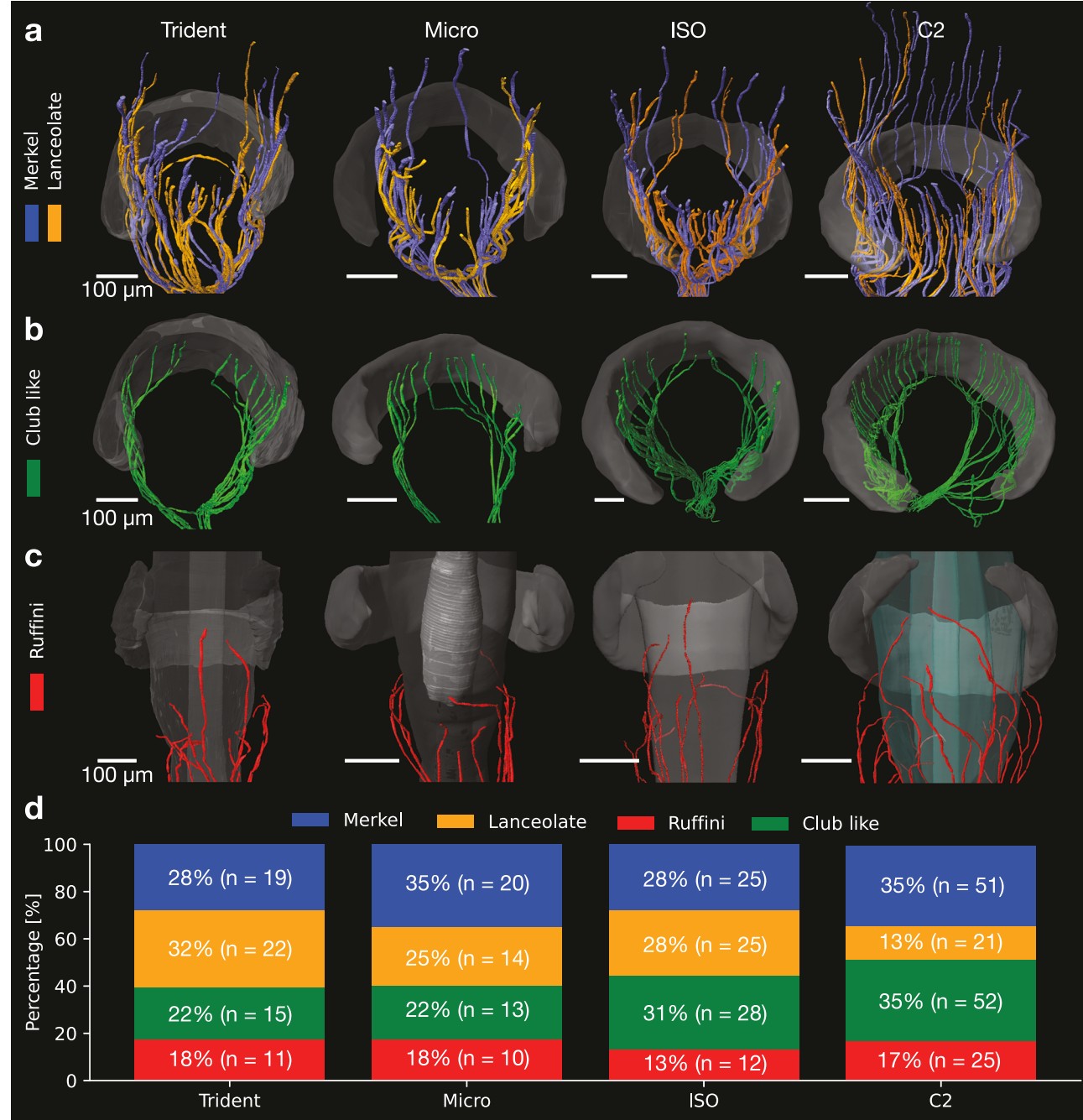

**Fig. 4 | Types and endings of myelinated afferents across different vibrissae.**
**a** Ring sinus-level Merkel- and lanceolate-endings (blue and orange respectively) across different vibrissae. **b** Club-like endings terminating adjacent to the ringwulst are shown in green. Not the relatively higher abundance of club-like endings and even tiling in the C2 follicle. **c** Below, ringwulst Ruffini-endings are shown in red. **d** Bar plot showing relative composition of afferent types across different vibrissa follicles.

## Sample embedding

Stained follicle samples were dehydrated in an ascending ethanol series of 25/50/70% Ethanol in H$_2$O for 2 h per step and left in 70% ethanol overnight at room temperature. Subsequently, samples were placed in tissue cassettes and placed in the Hypercenter XP Tissue Processing system for paraffin infiltration. The program set for paraffinization runs through an ascending ethanol (80/92/96/96/96/96/100) series, a 100% isopropanol, and three 100% Rotinistol, all at 39 °C, and finally three 100% paraffin steps at 60 °C. All steps run under applied vacuum for homogeneous infiltration. Fully infiltrated paraffin samples were then placed in 200 μm pipette tips and carefully filled with liquid paraffin on a Leica EG1160 tissue embedding station.

The trident vibrissa follicle X-ray image was embedded in Epon epoxy resin (Sigma Aldrich) following standard protocols. In short: the tissue was first dehydrated in an ascending ethanol series (50/75/100%) and Aceton (50/75/100%), followed by infiltration with Epon resin (25/50/75/100%). All steps were performed at room temperature. Finally, the sample was cured in a fresh 100% Epon mixture in a 1.5 mm diameter Kapton polyimide tube at 60 °C for 48 h.

## X-ray phase-contrast tomography

Stained and unstained paraffin-embedded vibrissa follicles were haltered vertically in 200 μm pipette tips filled with paraffin. Volumetric data was acquired by propagation-based X-ray phase-contrast tomography with an

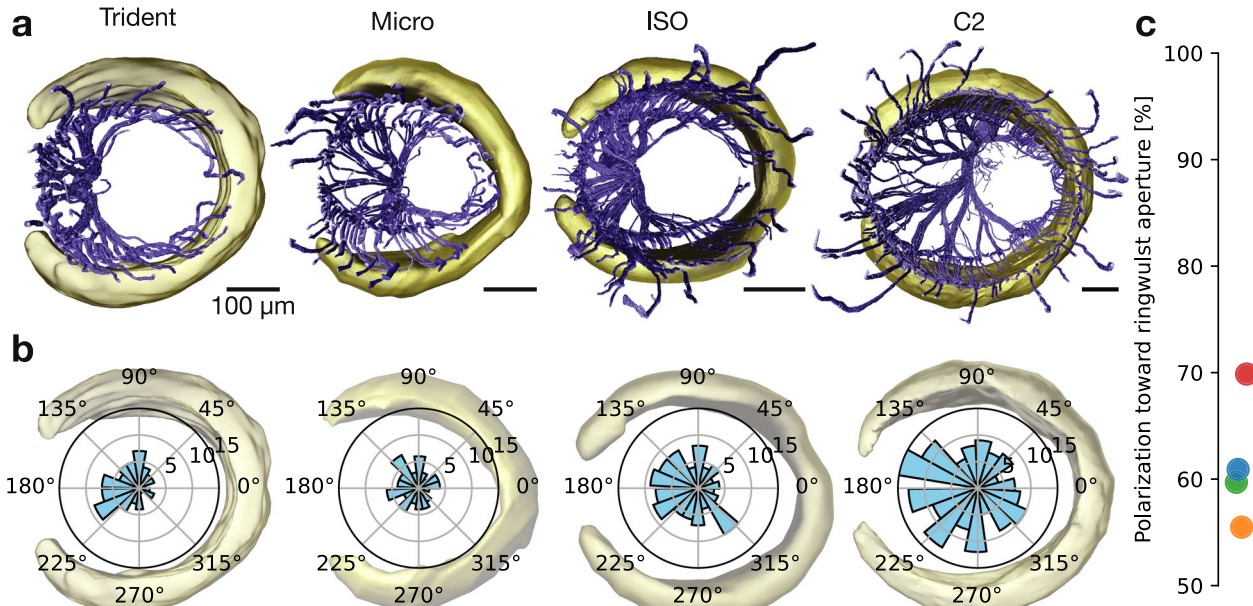

**Fig. 5 | Polarization of afferent innervation differs across vibrissae. a** Volume rendering showing asymmetric innervation (blue) and ringwulst (yellow) across different vibrissae as indicated above. **b** Polar histograms showing innervation polarization toward the ringwulst aperture across different vibrissae. **c** Polarization index measured by the relative amount of afferents (%) on the ringwulst aperture side. Trident in red, micro in green, lSO in blue, and C2 in orange.

unfocused, quasi-parallel synchrotron beam (PB) at the GINIX endstation with a photon energy $E_{ph}$ of 13.8 keV, selected by a Si(111) monochromator. Projections were recorded using a microscope detection system (Optique Peter, France) with a 50-µm-thick LuAG: Ce scintillator and a 10× magnifying microscope objective onto a sCMOS sensor (pco.edge 5.5, PCO, Germany). This configuration enables a field of view (FOV) of $1.6 \times 1.4$ mm for each projection, sampled at a pixel size of 650 nm. The continuous scan mode of the setup allows the acquisition of a tomographic recording with 3000 projections over 360° in less than 2 min. After recording the tomographic dataset, dark field and flat field images were acquired. For each follicle, three tomographies with an overlap of ~20% were acquired.

### Phase retrieval and tomographic reconstruction
First, the raw detector images were corrected by dark subtraction and empty beam division. Phase retrieval was performed for each projection, using the linear CTF approach[34,35] implemented in the HoloTomoToolbox[36]. Apart from phase retrieval, the HoloTomoToolbox provides auxiliary functions, which help to refine the Fresnel number or to identify the tilt and shift of the axis of rotation[36]. Tomographic reconstruction of the datasets was performed by the ASTRA toolbox[37,38], using the iradon-function and a Ram-Lak filter.

### Image post-processing
Follicles were covered in two to three imaging volumes, which were subsequently stitched using the FIJI pairwise stitching plugin[39] or with the NRStitcher[40]. After importing the stitched follicle datasets into Amira, we cropped it to the outer follicle boundaries, which resulted in the depicted volume images in Fig. 1.

Tomographic images were segmented in an extended version of the Amira software (AmiraZIBEdition 2022.17, Zuse Institute Berlin, Germany). A combination of the "lasso" and "brush" tools was used to manually label axons and follicle structures. Labels were placed every 5–50 images and interpolated in between.

### Statistics and reproducibility
Data was analyzed and plotted using the matplotlib and seaborn plugins for Python 3.6 (https://www.python.org). 3D reconstructions were visualized in Amira (Figs. 1–3). Figure layouts were prepared with Adobe Illustrator (version 28.1).

Myelinated axon counts in Fig. 3 were obtained from vibrissa follicle X-ray images and represent $n = 3$ in all cases. These counts either stem from a dense segmentation or from a cross-section count through the vibrissal nerve.

We quantified fiber diameter by averaging the fiber cross sections of axons from their entry point into the follicle until follicle mid-height, where no endings had yet formed. We observed no major diameter variations per axon in regions with myelination. In our analysis, we did not consider distal regions, where myelination tends to become smaller.

Midpoints of the axon terminals were noted as xy-coordinates within the image data and centered, with the vibrissal shaft midpoint at ringwulst level as $xy = (0, 0)$ coordinate. Next, we computed the slope of a line with the terminal $xy$ position as $x1y1$ and the vibrissal shaft as $x2y2$ coordinate. From the slope we calculated the angle by taking the arctan of the slope.

After performing a one-way ANOVA, post-hoc Dunn's pairwise comparison for comparison of axon counts (Fig. 3c) and fiber diameter (Fig. 3d) was performed using the scikit_posthocs functions. Linear regression for axon area–terminal height dependency was calculated using the scipy linregress function.

### Afferents exclusion criteria
For afferent classifications, we excluded all axons that we lost during segmentation within their axon arms. Axons that showed a clear trajectory and left the axon arms, or when in direct proximity to the root sheath, were included in the analysis, since we presumed those axons to be close to their terminal region. We justify our exclusion criteria as follows: (1) Axon trajectories are mostly curved while within axon arms. (2) Axons that leave the axon arms show a clear trajectory toward the shaft, which indicates their terminal territory. (3) As soon as macroscopic arms disappear and axons run individually, their trajectories are mostly straight, and angular territories don't change much. Thus, our exclusion criteria reduce our axon data to those axons that are at or close to their terminal regions. Further, we excluded axon fragments, which could not be connected to a "ground-cable" from our axon-based analysis, since we could not sufficiently exclude them as mere branches of another axon. Axon paths that start 100 µm below the

ringwulst were considered as "ground-cables," as branching only occurred upwards of this position.

## Afferent type classification

We classified afferent types based on a multidimensional scale, which comprised vertical ending territory (divided into ring sinus-, ringwulst-, and cavernous sinus-level territories) and ending morphologies (visible root sheath entrance, glassy membrane adjacency, branching, multi-branching (>2 branches), and trajectories toward the shaft). Based on these criteria, afferent types were classified by their dominating characteristics as: Merkel-afferents (ring sinus-level terminal, root sheath entrance), lanceolate-afferents (ring sinus-level terminal, on glassy membrane terminal), club-like-afferents (ringwulst terminal, on glassy membrane terminal), and Ruffini-like afferents (below ringwulst terminal, visible trajectories toward the root sheath). In our classification matrix, we only included those afferents that are connected to a "ground-cable".

## Reporting summary

Further information on research design is available in the Nature Portfolio Reporting Summary linked to this article.

## Data availability

Data related to the manuscript can be accessed via the G-Node repository https://doi.org/10.12751/g-node.gws6k9. Access to uncompressed imaging data and our derived segmentation will be provided upon request by the corresponding author (michael.brecht@bccn-berlin.de). The source data behind the graphs in the paper can be found in Supplementary Data 1.

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

## Acknowledgements

We thank Andreea Neukirchner, Undine Schneeweiß, Konstantin Strauss, Lennart Eigen, Matias Mugnaini and Markus Osterhoff. B.G. and M.B. are supported by BCCN Berlin, Humboldt-Universität zu Berlin, and the Deutsche Forschungsgemeinschaft (DFG, German Research Foundation) under Germany´s Excellence Strategy—EXC-2049—390688087. T.S. and J.R. receive funding from SFB1456/A3 Mathematics of Experiment. We acknowledge DESY (Hamburg, Germany), a member of the Helmholtz Association HGF, for the provision of experimental facilities. Parts of this research were carried out at PETRA III, and we would like to thank Fabian Westermeyer for assistance in using the P10 GINIX endstation. Beamtime was allocated for proposals I-20220980 and I-20240117. B.G. is supported by Wübben Foundation Wissenschaft and Max Planck Society. We acknowledge support by the Open Access Publication Fund of Humboldt-Universität zu Berlin.

## Author contributions

Conceptualization: B.G., T.R. and M.B.; methodology and materials: B.G., T.R., J.A., J.R., T.S. and M.B.; investigation: B.G., T.R. and M.B.; formal analysis: B.G., T.R. and M.B.; writing: B.G., T.R., J.R., T.S. and M.B.; supervision: T.S. and M.B.; funding acquisition: T.S. and M.B.

## Funding

## Competing interests

The authors declare no competing interests.
