## [Transparent Peer Review file · Communications Biology]

Follicle architecture and innervation of functionally distinct rat vibrissae

Corresponding Author: Dr Michael Brecht

Version 0:

Reviewer comments:

Reviewer #1

(Remarks to the Author)

A beautiful anatomy paper, building on the work of the group to compare the anatomy of different vibrissal follicles. Findings reveal that they actually have a very similar anatomy. However, notable differences could be observed in afferent arrangement and follicle size. The imaging work here is absolutely beautiful, and could be emphasized (“bigged up”) a little more in the abstract and introduction.

Introduction:

1. A bit of tense-switching in the introduction, makes it a little clunky. i.e. majority is past tense, as is usual, but check i.e. “Decisive functional information about vibrissal innervation came from afferent recording and labeling experiments”
2. The wording is a little subjective, please can you remove terms such as trail-blazing and ground-breaking.
3. As well as the walrus study, similar results have been observed on harbour seals (i.e. Grant et al. 2013), and it might be worth doing a quick search to see if there are more examples.
4. I was just thinking as I was reading... do the more ventral whiskers also sense the ground, and perhaps ground speed, as they often contact the floor? i.e. Arkley et al. 2014. And in a similar vein, can macrovibrissae do wind sensing too, as well as the supraorbital? I know it wouldn't be C2 specifically, but perhaps some of the other macrovibrissae? It might be nice to state this somewhere in the manuscript to really outline our full understanding of functional differences from the start – what is known as well as what is not known yet.

Methods:

5. Some sections are a little short, perhaps some of the paragraphs can be combined?
6. Can you say what each of the stains do?
7. Please can you add more details to the statistics section. What were the tests for? What was the main test before the posthoc test? Which variables went in? State dependent, independent etc.

Results:

Lovely, well done!

8. Would it be helpful to normalise all the measurements to the size of the different follicles to make comparisons easier? Perhaps comment on your decision for this in the methods or discussion sections?

Discussion:

9. For vibrissal hair length, can you discuss what the lengths might mean. i.e. Dougill et al. (2020 and 2022) said that longer whiskers might be more flexible – which might be useful for a wind sensor? Whereas perhaps something that contacts more often might be stiffer?
10. “As earlier authors we found a similarity of accessory structures” this does not really make sense. Perhaps “in agreement with previous authors...”?
11. You conclude with the syncatron imaging, which is absolutely fantastic. I wonder whether you can make more of this a little higher up, perhaps even emphasizing the novelty? It would be nice to say the benefit of syncatron over micro-CT (i.e. Milne et al's recent work).

Figures:

12. I really like table 1, can we make the rat figures a little bigger so we can clearly see the whisker locations?
13. The segmentation images are all beautiful! Wow! What a lot of work, well done!
14. For figure 3a, what are the horizontal coloured lines? Would it be possible to show the outlines of the follicles here to show where these all sit in relation?

Reviewer #2

(Remarks to the Author)

This is an important follow-up study to a beautiful and astonishing paper using micro-X-ray CT (Gerhardt et al., 2025). Using a novel method for non-destructive microscopic observation of whole-mount preparations, they have provided a comprehensive innervation patterns of whisker vibrissae located at four different locations. This is a meticulous and painstaking work, and a commendable achievement. This information will be of great importance and interest, especially for researchers studying peripheral sensory nerves and cutaneous sensory receptors. This also certainly contribute to elucidating the central mechanisms of perception.

The reviewer has two major concerns and several minor comments:

1) This is incredibly valuable information obtained using an innovative technology, so it is important that the morphological analyses alone are carefully described and compared. The authors' conclusion; "We conclude that overall innervation pattern follows a common blue-print across follicles, while fiber diameter and exact termination patterns are follicle specific." (Line number is missing. The above L332, perhaps.) As stated below, it may be a bit problematic to draw this conclusion in the Results. The authors may discuss this in the Discussion section, though.

Citing previous papers on behavioral analysis, the authors assigned the four types of functional features (three and multiple) to the four whisker follicles distributed different areas. As the authors point out, the resolution of the innovative microscopic device is limited, so the morphological analysis of the terminal part cannot be performed sufficiently. Furthermore, without any additional new comprehensive evidence, e.g. from either molecular biology or electrophysiology, or without comparison more specifically and decently with previous studies, it seems not logical to link each whisker follicle to a specific behavioral function based only on macroscopic structural analysis of the hair follicle and morphological analysis of the thickness and course of the myelinated fiber bundles and surrounding apertures.

2) The Methods should be revised to make it easier to understand. C2 and the other three follicles might be not processed as the completely same methods according to the Methods. Only a(?) C2 vibrissa follicle was stained in 1% OsO4 (L717-8). Could this be the reason why only C2 has noticeably thinner myelinated fibers? The shrinkage rate of tissues changes when they are immersion fixed after being taken from unfixed cadavers, so measurements must be compared using specimens prepared using the same method for all tissues, or an annotation should be necessary.

It is not easy to understand how many whiskers were taken from how many cadavers, how the cadavers were taken/used/fixed, how many of each of the four types of whisker follicles, and which follicle of the Trident or ISO vibrissae were studied, and what method was used for individual follicles. These are necessary for highly informative morphological analysis like this study.

Minor comments.

- 1) Fig. 1b: It would be more helpful to clearly see the follicle size within each block, even with a drawing outline.
- 2) Table 1 : It would be helpful to include at least one site in the References for Macro vibrissa.
- 3) L110-111: Do the follicles used in this study keep azimuthal information?
- 4) Fig.2a: Do they have hair papillae?
- 5) Fig.2b,c: The top of the ring sinus of the ISO follicle might be missing during segmentation. The shadow around the inner conical body tells it. If not, what happened in the only ISO?
- 6) Fig.2b: Don't Micro and ISO follicles have sebaceous glands (green in Trident or C2)?
- 7) Fig.2, Fig.3: Please indicate superficial and deep vibrissal nerve bundles respectively.
- 8) L116-118: "cavernous sinus" should be added.
- 9) L122-124: There is a discrepancy to the Fig. 2d; the ringwulst of the Microvibrissa shows obviously smaller than the others.
- 10) L131: The term "axon arms" is still not popular, so that please put reference number of Gerhardt et al., 2025.

- 11) Fig.3f: Axon diameter  Myelinated axon diameter
- 12) L343-345: For classification of myelinated afferents, a reference (Gerhardt et al., 2025) is necessary.
- 13) L353-354: Fig.4c should be Fig. 4d?
- 14) L369-: The first paragraph is not Summary.
- 15) L379: "the functionally different vibrissae" is unclear. Behavioral function?
- 16) L399: It would be helpful to write "bauplan" together with "body plan".
- 17) L770 and L774: Amira software (.....) should be L770.
- 18) L793: arctan  arctangent
- 19) L810-: "Afferent exclusion criteria" and "Afferent type classification" are mostly the same as Gerhardt et al., 2005.

In addition, the reviewer concerns about parts of the Discussion (L405-408), even though any authors are known to be open to discussion. The authors hypothesize that that the results according to the differences of fiber diameter may reflect the need for temporal immediacy of detecting stimuli sampled through the Micro and Trident, while C2-touch might be comparably less timely relevant (L406-408). If the authors mention this, please provide any proper references. Each type of follicle should have a significant number of thick as well as fine fibers with receptors and each whisker, wherever, should have responded enough timely and sensitively, if only the hair shaft is in sufficient contact with an object.

Reviewer #3

(Remarks to the Author)

Rodent vibrissae are complicated somatosensory organs with dense innervation and unique accessory structures. The manuscript by Gerhardt, Rodde and colleagues provides a detailed characterization of the sensory complexes of functionally different vibrissae using synchrotron X ray imaging. They show that ringwulst and associated afferents maintain height and diameter despite vastly different hair sizes. In contrast, the different hairs vary in the density and polarization of afferents, suggesting functional specializations of innervation. Overall, this manuscript provides a detailed anatomical characterization of vibrissa innervation, strong quantification, and thoughtful speculations as to the functional basis for differences that are open to further study. This provides an important step in understanding functional organization of somatosensory end organs. A few clarifications and quantifications can help to strengthen this manuscript and assist with reproducibility.

Major:

1. One of the more surprising primary findings of the manuscript is that the ringwulst size is constant across different follicles, and that this is in keeping with the consistent diameter of hair follicles. Neither of these assertions are quantified in the manuscript. Given that the authors have the data to do these quantifications and the importance of the finding, these quantifications should be included.
2. In figure 4, qualitative observations are described of the Merkel and lanceolate endings that are quite difficult to see with the resolution of the renderings. It would be helpful to show higher resolution images of these endings with annotations.
3. In the discussion section labelled 'differences in innervation' the authors note the surprising finding about differences in hair follicle diameter between vibrissae. It is very interesting that the C2 has more myelinated fibers but smaller axon diameter. The authors suggest that this difference may be due to a need for immediacy of the signal in some vibrissae. An alternative hypothesis could be that, this difference could reflect molecular and functional sub-specializations of afferent types in different follicles with similar structures.

Minor:

1. Line 347- 'club like afferences' should be 'club-like afferents'
2. Lines 381-382- 'The most striking differences concern vibrissa hair length and correlated which correlates with follicle size'. I think there is a word missing, I don't understand what the authors are trying to convey.
3. Table 1: C2 references are just given as 'Many'. It is understood that this is a large list and the authors may not be able to reasonably provide all, but some references would be beneficial for a non-specialist.
4. Figure 3C- the authors should give some statistical comparison between groups to support their findings.
5. More details should be provided or how fiber diameter is quantified, as it can vary across the length of the axon.
6. Figure 4d. The authors should provide a sense for the variability in the data and whether groups are statistically different. Similarly, the point is made about the differences in density and tiling of club-like afferents in 4C, but no quantification is given for this.
7. Figure 5. Some measure of variability and statistical differences should be provided.

Version 1:

Reviewer comments:

Reviewer #1

(Remarks to the Author)

I am happy that the authors have taken the time to address my comments, thank you.

The paper that I suggested to include (sorry, I did not provide the whole reference, and the first author has had a surname change, which was totally my fault) was:

Elder, A., Evans, E., Brassey, C., Kitchener, A.C., Hantke, G. and Grant, R., 2024. Describing the musculature of mystacial pads in harbour seals (*Phoca vitulina*) using diceCT. *Journal of Anatomy*.

It might be nice to include as it is another 3D imaging paper - although of course with a different species and focus.

Reviewer #2

(Remarks to the Author)

The authors worked very hard, responded appropriately to the reviewers' comments, and carefully revised the manuscript. I think it is an easy-to-read, high-quality paper. I appreciate their efforts. The authors' revisions are mostly perfect except for the third major point of the Reviewer #3 which is the same of the additional comment at the bottom of this reviewer (#2).

The sentence in L241-244 are misleading to the readers. Corrections are needed. It is incorrect to classify the conduction velocity of each hair follicle as fast or slow. Every follicle contains many myelinated fibers of different diameters. Some are thick, A β -, and some are thin, A δ -fibers. These multiple fibers may fire under various conditions during exploration, and not all of them fire at the same time, so it is not reasonable to assess the conduction velocity of the entire hair follicle by the average value of the overall fiber diameter.

The followings are minor points.

The reviewer was not indicate that the CS should be reconstructed, but rather that it would be helpful to include the letters of RS and CS in the Figures. The same gose DVN and SVN.

By the way, the reviewer apologizes for confusing the author by using the word "azimuth" incorrectly in the sentence. The reviewer just wanted to confirm with the author whether they could compare the direction of the missing part of R_w of the body for each of the four hair follicles. This question does not affect the manuscript.

Reviewer #3

(Remarks to the Author)

All of my concerns have been addressed in the revised manuscript. The revisions will help with reproducibility and interpretation of the data and strengthen the paper overall. I stand by my original review that this manuscript provides important findings towards understanding functional organization of the somatosensory system.

**Follicle architecture and innervation of functionally distinct rat vibrissae
- Rebuttal letter round 1 -**

Dear Editors and Reviewers,

We are most thankful for the fair and constructive critic points regarding our manuscript. We also note that the overall assessment by the reviewers is positive and we are delighted about this feedback. All of the points raised have been addressed by us and can be summarized as follows:

- 1. Method details (statistics, imaging, staining) were added.**
- 2. Missing reconstruction details were added.**
- 3. We added citations.**
- 4. We improved the readability of the ms and implemented all referee suggestions point-by-point.**

We think these changes greatly improved our article. Below, we repeat the points by the referees in grey italics before responding to them point by point.

Reviewers' comments:

Reviewer #1 (Remarks to the Author):

A beautiful anatomy paper, building on the work of the group to compare the anatomy of different vibrissal follicles. Findings reveal that they actually have a very similar anatomy. However, notable differences could be observed in afferent arrangement and follicle size. The imaging work here is absolutely beautiful, and could be emphasized (“bigged up”) a little more in the abstract and introduction.

Comment: The referee summarizes and compliments our manuscript. Thank you.

Change: None

Introduction:

1. A bit of tense-switching in the introduction, makes it a little clunky. i.e. majority is past tense, as is usual, but check i.e. “Decisive functional information about vibrissal innervation came from afferent recording and labeling experiments”

Comment & Change: We reworked the wording of the manuscript to unify the tense in which it is written.

2. The wording is a little subjective, please can you remove terms such as trail-blazing and ground-breaking.

Comment: We agree.

Change: Such words have been removed and replaced by more neutral language.

3. As well as the walrus study, similar results have been observed on harbour seals (i.e. Grant et al. 2013), and it might be worth doing a quick search to see if there are more examples.

Comment: Thank you for noting this. The mentioned reference has already been included in the original manuscript (ref. 17, line 73 - 75), along the walrus study and other studies (ref. 14, ref. 15).

Change: None.

4. I was just thinking as I was reading... do the more ventral whiskers also sense the ground, and perhaps ground speed, as they often contact the floor? i.e. Arkley et al. 2014. And in a similar vein, can macrovibrissae do wind sensing too, as well as the supraorbital? I know it wouldn't be C2 specifically, but perhaps some of the other macrovibrissae? It might be nice to state this somewhere in the manuscript to really outline our full understanding of functional differences from the start - what is known as well as what is not known yet.

Comment: In general, relatively little is known about functional differences of vibrissae. The reason is that the vast majority of vibrissa-studies focused on the macrovibrissae.

Change: We extended the introduction of functional differences of vibrissae (line 79).

Methods:

5. *Some sections are a little short, perhaps some of the paragraphs can be combined?*

Comment & Change: We agree and changed this.

6. *Can you say what each of the stains do?*

Comment: Sure, we apologies for not being more specific with this in the original draft.

Change: We have added these explanations to the methods section to make it more accessible to non-expert readers.

7. *Please can you add more details to the statistics section. What were the tests for? What was the main test before the posthoc test? Which variables went in? State dependent, independent etc.*

Comment: The post-hoc test was performed for comparison of fiber diameter (Fig. 3d) and followed an one way ANOVA test.

Change: We updated the method section to clarify our statistical methods.

Results:

Lovely, well done!

8. *Would it be helpful to normalise all the measurements to the size of the different follicles to make comparisons easier? Perhaps comment on your decision for this in the methods or discussion sections?*

Comment: Thank you. We agree that normalizing measures to follicle size helps interpreting the data. We have already done this in e.g. Figure 2c, 3e and 3f but have not communicated this explicitly except for in the figure. We carefully thought about other analysis where normalization to follicle size could be beneficial (Figure 4 and 5), but came to the conclusion, that it is not a useful measure there, because either percentages are communicated (Figure 4) or distributions of afferents around the follicle (Figure 5).

Change: We changed the results paragraph to draw more attention to the normalization that has been performed.

Discussion:

9. *For vibrissal hair length, can you discuss what the lengths might mean. i.e. Dougill et al. (2020 and 2022) said that longer whiskers might be more flexible - which might be useful for a wind sensor? Whereas perhaps something that contacts more often might be stiffer?*

Comment: We agree.

Change: We now include this in the discussion.

10. *“As earlier authors we found a similarity of accessory structures” this does not really make sense. Perhaps “in agreement with previous authors...”?*

Comment & Change: We agree and have updated the wording.

11. *You conclude with the synchrotron imaging, which is absolutely fantastic. I wonder whether*

you can make more of this a little higher up, perhaps even emphasizing the novelty? It would be nice to say the benefit of synchrotron over micro-CT (i.e. Milne et al's recent work).

Comment: We agree – synchrotron X-ray imaging indeed provides a significantly different perspective on sensory architecture than conventional microCT imaging.

Change: We now further elaborate on this as suggested. However, we could not find the mentioned reference.

Figures:

12. I really like table 1, can we make the rat figures a little bigger so we can clearly see the whisker locations?

Comment & Change: We updated the table accordingly.

13. The segmentation images are all beautiful! Wow! What a lot of work, well done!

Comment: Thank you!

Change: None.

14. For figure 3a, what are the horizontal coloured lines? Would it be possible to show the outlines of the follicles here to show where these all sit in relation?

Comment: The horizontal lines indicate the plane of the nerve crosssections shown in Figure 3b. Since we already show innervation in context with the follicle in Figure 2a, we wanted to refrain from showing the follicle outlines again, however.

Change: We now note this in the figure legend.

Reviewer #2:

This is an important follow-up study to a beautiful and astonishing paper using micro-X-ray CT (Gerhardt et al., 2025). Using a novel method for non-destructive microscopic observation of whole-mount preparations, they have provided a comprehensive innervation patterns of whisker vibrissae located at four different locations. This is a meticulous and painstaking work, and a commendable achievement. This information will be of great importance and interest, especially for researchers studying peripheral sensory nerves and cutaneous sensory receptors. This also certainly contribute to elucidating the central mechanisms of perception.

Comment: The referee summarizes our paper and puts the finding into a broader perspective. We thank the referee for the enthusiastic assessment of our work.

Change: None.

The reviewer has two major concerns and several minor comments:

1) This is incredibly valuable information obtained using an innovative technology, so it is important that the morphological analyses alone are carefully described and compared. The authors' conclusion; "We conclude that overall innervation pattern follows a common blueprint across follicles, while fiber diameter and exact termination patterns are follicle specific." (Line number is missing. The above L332, perhaps.) As stated below, it may be a bit problematic to draw this conclusion in the Results. The authors may discuss this in the Discussion section, though.

Comment: We agree with the referee.

Change: We now limit our interpretations solely to the discussion section of the manuscript.

Citing previous papers on behavioral analysis, the authors assigned the four types of functional features (three and multiple) to the four whisker follicles distributed different areas. As the authors point out, the resolution of the innovative microscopic device is limited, so the morphological analysis of the terminal part cannot be performed sufficiently. Furthermore, without any additional new comprehensive evidence, e.g. from either molecular biology or electrophysiology, or without comparison more specifically and decently with previous studies, it seems not logical to link each whisker follicle to a specific behavioral function based only on macroscopic structural analysis of the hair follicle and morphological analysis of the thickness and course of the myelinated fiber bundles and surrounding apertures.

Comment: The referee correctly notes that the evidence on the functional role of different whiskers is rather limited.

Change: We added the following qualifier: **Note, that the behavioral evidence for specific vibrissae functions is rather limited, and we view all afore-mentioned functional assignments of the different vibrissae more as working hypothesis than ultimate functional classifications (line 107).**

2) The Methods should be revised to make it easier to understand. C2 and the other three follicles might be not processed as the completely same methods according to the Methods. Only a(?) C2 vibrissa follicle was stained in 1% OsO4 (L717-8). Could this be the reason why

only C2 has noticeably thinner myelinated fibers? The shrinkage rate of tissues changes when they are immersion fixed after being taken from unfixed cadavers, so measurements must be compared using specimens prepared using the same method for all tissues, or an annotation should be necessary.

Comment: We agree with the referee that the methods regarding our sample preparations lack clarity. Indeed, all samples have been stained with osmium (which is responsible for staining membranes and myelin sheaths) and hence we reckon that fiber diameters should be comparable across samples. Both microvibrissae and the large supraorbital vibrissa follicles have been stained with uranyl acetate in addition, which mainly contributes to enhancing contrast of intracellular features.

Change: We added a table in the method section which lists the exact staining procedure for each sample. We also added explanation for what each staining results in as suggested by referee #1.

It is not easy to understand how many whiskers were taken from how many cadavers, how the cadavers were taken/used/fixed, how many of each of the four types of whisker follicles, and which follicle of the Trident or ISO vibrissae were studied, and what method was used for individual follicles. These are necessary for highly informative morphological analysis like this study.

Comment: This partially relates to our answer above.

Change: We incorporated the information about the rat cadaver which each sample originates from into a table in the methods section.

Minor comments.

1)Fig. 1b: It would be more helpful to clearly see the follicle size within each block, even with a drawing outline.

Comment & Change: We added this detail to Figure 1.

2)Table 1:It would be helpful to include at least one site in the References for Macro vibrissa.

Comment & Change: We agree and have added a list of references to the table for the C2 vibrissa follicle.

3)L110-111: Do the follicles used in this study keep azimuthal information?

Comment: We are not quite sure, what the referee is asking here. We know from the position of the ringwulst opening, the orientation of each whisker *in situ*.

Change: None.

4)Fig.2a: Do they have hair papillae?

Comment: Yes the follicles do have hair papillae, however the transition from papillae to hair is smooth and was not specifically highlighted with our stained which aimed at increasing

contrast of the myelinated innervation. Hence, we could not distinguish it in a consistent manner and chose not to show it in the present work.

Change: None.

5) *Fig.2b,c: The top of the ring sinus of the ISO follicle might be missing during segmentation. The shadow around the inner conical body tells it. If not, what happened in the only ISO?*

Comment: Indeed, this missing detail is a reconstruction artifact and does not reflect a difference in follicle architecture.

Change: We revisited the segmentation of the vibrissa follicle and improved on this detail.

6) *Fig.2b: Don't Micro and ISO follicles have sebaceous glands (green in Trident or C2)?*

Comment: Same as above.

Change: We added sebaceous glands to the segmentation.

7) *Fig.2, Fig.3: Please indicate superficial and deep vibrissal nerve bundles respectively.*

Comment: Thanks for noting this.

Change: We added this description.

8) *L116-118: "cavernous sinus" should be added.*

Comment: Segmentation of the cavernous sinus is complicated because of its intricate structure with skins and trabeculae. Both trabeculae and tissue sheaths are difficult to segment in a consistent manner across the very different sized vibrissa follicles included in our paper because of varying shape and contrast. At present we do not have a better rendering of this structure than what is already inside of our paper. Rather than showing partially correct cavernous structure we would prefer leaving the paper as is.

Change: None.

9) *L122-124: There is a discrepancy to the Fig. 2d; the ringwulst of the Microvibrissa shows obviously smaller than the others.*

Comment: Here, we respectfully disagree with the referee. We do not say that the ringwulst has the exactly same size across vibrissae. Instead, we use the term 'near constant' and the qualifier 'near' implies minor differences. It remains true that ringwulst size is rather similar across follicle of very different size.

Change: None.

10) *L131: The term "axon arms" is still not popular, so that please put reference number of Gerhardt et al., 2025.*

Comment: We agree.

Change: We have added the reference in question.

11) *Fig.3f: Axon diameter  Myelinated axon diameter*

Comment: We agree that using the term axon diameter is inappropriate here.

Change: We changed it to fiber diameter in line with our wording elsewhere in the manuscript.

12) L343-345: *For classification of myelinated afferents, a reference (Gerhardt et al., 2025) is necessary.*

Comment: We agree.

Change: We added the reference in question.

13) L353-354: *Fig.4c should be Fig. 4d?*

Comment & Change: Yes, we changed the text accordingly.

14) L369-: *The first paragraph is not Summary.*

Comment: Ok.

Change: We removed this heading.

15) L379: *“the functionally different vibrissae” is unclear. Behavioral function?*

Comment: By vibrissa function we mean the type of sensory information that can be sampled with it during behavior.

Change: We have now clarified that in the text.

16) L399: *It would be helpful to write "bauplan" together with "body plan".*

Comment: We appologies for using this german term without further explanation. We suggest that the word ‘bauplan’ can best be accompanied by the work ‘blue print’ for improved understandability (rather than body plan).

Change: We now write ‘bauplan’ together with ‘blue print’

17) L770 and L774: *Amira software (.....) should be L770.*

Comment & Change: We changed this.

18) L793: *arctan  arctangent*

Comment & Change: We changed this.

19) L810-: *“Afferent exclusion criteria” and “Affernt type classification” are mostly the same as Gerhardt et al., 2005.*

Comment: We agree.

Change: None

In addition, the reviewer concerns about parts of the Discussion (L405-408), even though any authors are known to be open to discussion. The authors hypothesize that that the results according to the differences of fiber diameter may reflect the need for temporal immediacy of detecting stimuli sampled through the Micro and Trident, while C2-touch might be comparably less timely relevant (L406-408). If the authors mention this, please provide any proper references. Each type of follicle should have a significant number of thick as well as fine fibers with receptors and each whisker, wherever, should have responded enough timely and sensitively, if only the hair shaft is in sufficient contact with an object.

Comment: We agree with the referee's reasoning.

Change: We extended the interpretation of these findings along the lines suggested by referee 2. Please see point 3 of our response to referee 2.

Reviewer #3:

Rodent vibrissae are complicated somatosensory organs with dense innervation and unique accessory structures. The manuscript by Gerhardt, Rodde and colleagues provides a detailed characterization of the sensory complexes of functionally different vibrissae using synchrotron X ray imaging. They show that ringwulst and associated afferents maintain height and diameter despite vastly different hair sizes. In contrast, the different hairs vary in the density and polarization of afferents, suggesting functional specializations of innervation. Overall, this manuscript provides a detailed anatomical characterization of vibrissa innervation, strong quantification, and thoughtful speculations as to the functional basis for differences that are open to further study. This provides an important step in understanding functional organization of somatosensory end organs. A few clarifications and quantifications can help to strengthen this manuscript and assist with reproducibility.

Comment: The referee provides a detailed summary of our manuscript and comes to a very positive assessment – thank you.

Change: None.

Major:

1. One of the more surprising primary findings of the manuscript is that the ringwulst size is constant across different follicles, and that this is in keeping with the consistent diameter of hair follicles. Neither of these assertions are quantified in the manuscript. Given that the authors have the data to do these quantifications and the importance of the finding, these quantifications should be included.

Comment: We agree.

Change: We now provide a quantitative take on our previously verbally communicated findings.

2. In figure 4, qualitative observations are described of the Merkel and lanceolate endings that are quite difficult to see with the resolution of the renderings. It would be helpful to show higher resolution images of these endings with annotations.

Comment: We agree with the referee.

Change: We added higher magnification images of endings as Supplementary Figure 1.

3. In the discussion section labelled 'differences in innervation' the authors note the surprising finding about differences in hair follicle diameter between vibrissae. It is very interesting that the C2 has more myelinated fibers but smaller axon diameter. The authors suggest that this difference may be due to a need for immediacy of the signal in some vibrissae. An alternative hypothesis could be that, this difference could reflect molecular and functional sub-specializations of afferent types in different follicles with similar structures.

Comment: We agree. We add that we do not yet fully understand the implications of different axon diameters across follicles.

Change: We now mention the referee's alternative hypothesis in the text.

Minor:

1. Line 347- 'club like afferences' should be 'club-like afferents'

Comment: We agree.

Change: We corrected this.

2. Lines 381-382- 'The most striking differences concern vibrissa hair length and correlated which correlates with follicle size'. I think there is a word missing, I don't understand what the authors are trying to convey.

Comment: We agree.

Change: It was supposed to say 'The most striking differences concern vibrissa hair length which correlates with follicle size'. We changed the text accordingly.

3. Table 1: C2 references are just given as 'Many'. It is understood that this is a large list and the authors may not be able to reasonably provide all, but some references would be beneficial for a non-specialist.

Comment: We agree.

Change: We added the most appropriate papers as citations to the table.

4. Figure 3C- the authors should give some statistical comparison between groups to support their findings.

Comment: We agree.

Change: We added a statistical comparison (one way ANOVA) to compare the axon counts across the different follicles.

5. More details should be provided on how fiber diameter is quantified, as it can vary across the length of the axon.

Comment: We agree.

Change: We added a sentence to the relevant methods section.

6. Figure 4d. The authors should provide a sense for the variability in the data and whether groups are statistically different. Similarly, the point is made about the differences in density and tiling of club-like afferents in 4C, but no quantification is given for this.

Comment: We do not have a satisfactory answer here, because our analysis is based on few (1-3) follicles per group. It appears that follicle differences hold across this sample, but we are far from statistically sound assessment. Our assessment of club-like tiling differences relied on visual inspection and could be confirmed through radial histograms (Referee Figure 1, below).

Change: None.

Referee Figure 1 Tiling of club-like afferents across vibrissae. From left to right: Histogram of club like afferences of the trident, micro, ISO and C2 vibrissae follicle. 0° = ringwulst midpoint, 180° = ringwulst aperture.

7. Figure 5. Some measure of variability and statistical differences should be provided.

Comment: Please see our answer above. Specifically, we included measures of variability where possible. Given the timely analysis of such datasets, we could not provide accurate estimates for variability of all measures that we describe. In our first paper on three-dimensional vibrissa follicle innervation we concluded that variability of this tactile sensor is small (Gerhardt et al., 2025). Here, it was our goal to describe major anatomical differences of the vibrissae under investigation and reckon that assessing variability of vibrissa follicle innervation lies beyond the scope of this current manuscript.

Change: None.

This is an important follow-up study to a beautiful and astonishing paper using micro-X-ray CT (Gerhardt et al., 2025). Using a novel method for non-destructive microscopic observation of whole-mount preparations, they have provided a comprehensive innervation patterns of whisker vibrissae located at four different locations. This is a meticulous and painstaking work, and a commendable achievement. This information will be of great importance and interest, especially for researchers studying peripheral sensory nerves and cutaneous sensory receptors. This also certainly contribute to elucidating the central mechanisms of perception.

The reviewer has two major concerns and several minor comments:

- 1) This is incredibly valuable information obtained using an innovative technology, so it is important that the morphological analyses alone are carefully described and compared. The authors' conclusion; "We conclude that overall innervation pattern follows a common blue-print across follicles, while fiber diameter and exact termination patterns are follicle specific." (Line number is missing. The above L332, perhaps.) As stated below, it may be a bit problematic to draw this conclusion in the Results. The authors may discuss this in the Discussion section, though.

Citing previous papers on behavioral analysis, the authors assigned the four types of functional features (three and multiple) to the four whisker follicles distributed different areas. As the authors point out, the resolution of the innovative microscopic device is limited, so the morphological analysis of the terminal part cannot be performed

sufficiently. Furthermore, without any additional new comprehensive evidence, e.g. from either molecular biology or electrophysiology, or without comparison more specifically and decently with previous studies, it seems not logical to link each whisker follicle to a specific behavioral function based only on macroscopic structural analysis of the hair follicle and morphological analysis of the thickness and course of the myelinated fiber bundles and surrounding apertures.

- 2) The Methods should be revised to make it easier to understand. C2 and the other three follicles might be not processed as the completely same methods according to the Methods. Only a(?) C2 vibrissa follicle was stained in 1% OsO₄ (L717-8). Could this be the reason why only C2 has noticeably thinner myelinated fibers? The shrinkage rate of tissues changes when they are immersion fixed after being taken from unfixed cadavers, so measurements must be compared using specimens prepared using the same method for all tissues, or an annotation should be necessary.

It is not easy to understand how many whiskers were taken from how many cadavers, how the cadavers were taken/used/fixed, how many of each of the four types of whisker follicles, and which follicle of the Trident or ISO vibrissae were studied, and what method was used for individual follicles. These are necessary for highly informative morphological analysis like this study.

Minor comments.

- 1) Fig. 1b: It would be more helpful to clearly see the follicle size within each block, even with a drawing outline.
- 2) Table 1 : It would be helpful to include at least one site in the References for Macro vibrissa.
- 3) L110-111: Do the follicles used in this study keep azimuthal information?
- 4) Fig.2a: Do they have hair papillae?
- 5) Fig.2b,c: The top of the ring sinus of the ISO follicle might be missing during segmentation. The shadow around the inner conical body tells it. If not, what happened in the only ISO?
- 6) Fig.2b: Don't Micro and ISO follicles have sebaceous glands (green in Trident or C2)?
- 7) Fig.2, Fig.3: Please indicate superficial and deep vibrissal nerve bundles respectively.
- 8) L116-118: "cavernous sinus" should be added.
- 9) L122-124: There is a discrepancy to the Fig. 2d; the ringwulst of the Microvibrissa shows obviously smaller than the others.
- 10) L131: The term "axon arms" is still not popular, so that please put reference number of Gerhardt et al., 2025.
- 11) Fig.3f: Axon diameter → Myelinated axon diameter
- 12) L343-345: For classification of myelinated afferents, a reference (Gerhardt et al., 2025) is necessary.

- 13) L353-354: Fig.4c should be Fig. 4d?
- 14) L369-: The first paragraph is not Summary.
- 15) L379: "the functionally different vibrissae" is unclear. Behavioral function?
- 16) L399: It would be helpful to write "*bauplan*" together with "body plan".
- 17) L770 and L774: Amira software (.....) should be L770.
- 18) L793: arctan → arctangent
- 19) L810-: "Afferent exclusion criteria" and "Afferent type classification" are mostly the same as Gerhardt et al., 2005.

In addition, the reviewer concerns about parts of the Discussion (L405-408), even though any authors are known to be open to discussion. The authors hypothesize that that the results according to the differences of fiber diameter may reflect the need for temporal immediacy of detecting stimuli sampled through the Micro and Trident, while C2-touch might be comparably less timely relevant (L406-408). If the authors mention this, please provide any proper references. Each type of follicle should have a significant number of thick as well as fine fibers with receptors and each whisker, wherever, should have responded enough timely and sensitively, if only the hair shaft is in sufficient contact with an object.